# Current and Emerging Inhaled Antibiotics for Chronic Pulmonary *Pseudomonas aeruginosa* and *Staphylococcus aureus* Infections in Cystic Fibrosis

**DOI:** 10.3390/antibiotics12030484

**Published:** 2023-02-28

**Authors:** Danni Li, Elena K. Schneider-Futschik

**Affiliations:** Department of Biochemistry & Pharmacology, School of Biomedical Sciences, Faculty of Medicine, Dentistry and Health Sciences, The University of Melbourne, Parkville, VIC 3010, Australia

**Keywords:** cystic fibrosis, inhaled antibiotics, *P. aeruginosa*, *S. aureus*, chronic pulmonary infections

## Abstract

Characterized by impaired mucus transport and subsequent enhanced colonization of bacteria, pulmonary infection causes major morbidity and mortality in patients with cystic fibrosis (CF). *Pseudomonas aeruginosa* (*P. aeruginosa*) and *Staphylococcus aureus* (*S. aureus*) are the two most common types of bacteria detected in CF lungs, which undergo multiple adaptational mechanisms such as biofilm formation resulting in chronic pulmonary infections. With the advantages of greater airway concentration and minimized systemic toxicity, inhaled antibiotics are introduced to treat chronic pulmonary infection in CF. Inhaled tobramycin, aztreonam, levofloxacin, and colistin are the four most common discussed inhaled antibiotics targeting *P. aeruginosa*. Additionally, inhaled liposomal amikacin and murepavadin are also in development. This review will discuss the virulence factors and adaptational mechanisms of *P. aeruginosa* and *S. aureus* in CF. The mechanism of action, efficacy and safety, current status, and indications of corresponding inhaled antibiotics will be summarized. Combination therapy and the strategies to select an optimal inhaled antibiotic protocol will also be discussed.

## 1. Introduction

Cystic fibrosis (CF) is an autosomal recessive disease caused by mutations in a gene encoding a cystic fibrosis transmembrane conductance regulator (CFTR) protein [1]. CF prevalence reaches 1 in 2000 to 3000 live births in Caucasian populations, and the median-predicted survival time reached 65.6 years in 2021 [2,3]. It is a multisystemic disease causing many symptoms, including cough, repeated lung infections, pancreatic insufficiency, intestinal obstruction, failure of thrive, female subfertility and male infertility [1]. Different management strategies are required to relieve symptoms including antibiotics, nutritional supplement and physiotherapy [4]. Recently, highly effective CFTR modulators became available on the market, which are small molecules to restore the CFTR functions and address the underlying causative factors [5].

As the leading cause of morbidity and mortality in CF, respiratory symptoms are mainly driven by defective or deficient CFTR and the following pulmonary infections. In CF, the secretion of Cl- and HCO_3_− is disrupted by defective CFTR, resulting in ion accumulation and high intracellular osmotic pressure causing mucus dehydration, which promotes mucus tethering and hinders cilia beating (Figure 1) [6]. Impaired mucociliary clearance fails to clear pathogens and facilitates bacteria colonization in mucus plaques, among which *Pseudomonas aeruginosa* (*P. aeruginosa*) and *Staphylococcus aureus* (*S. aureus*) are the two most common types leading to acute and chronic pulmonary infections [1,7,8]. Inflammatory responses triggered by infection, damage pulmonary structures, resulting in declined lung function, pulmonary exacerbations, bronchiectasis, pulmonary insufficiency, and eventually respiratory failure. Due to ineffective early eradication therapies, pathogens will persist in the lungs causing chronic infections. Hence, effective antibiotics are desperately needed. Compared with a systemic regimen, inhaled antibiotics have the advantages of limited toxicity and greater airway concentrations [9]. As a result, many antibiotics were produced as inhalable formulations, such as tobramycin, aztreonam, and colistin, to treat different chronic pulmonary infections. However, for most of these nebulized drugs, data regarding their efficacy, and safety in treating CF patients is very limited. Consequently, an overview of the currently available inhaled antibiotics is urgently required for both professionals and CF patients to select the optimal treatment strategy of chronic pulmonary infections. This review will focus on the virulence factors and adaptation mechanisms of *P. aeruginosa* and *S. aureus* causing chronic pulmonary infections in CF patients and the available inhaled antibiotic therapies. Factors determining optimal inhaled antibiotic selection will also be discussed.

## 2. Major Colonized Pathogens in Cystic Fibrosis and the Adaptation to Chronic Infection

### 2.1. Pseudomona Aeruginosa

#### 2.1.1. Prevalence

Originally derived from the natural environment, the acquisition of Gram-negative, aerobic, rod-shaped bacterium *P. aeruginosa* in CF patients starts early and dominates from the second decade to the end of life, and is isolated from 55% adult patients [10,11,12,13]. Its prevalence continues to decrease in the 21st century, which is thought to be associated with early eradication therapies [3].

#### 2.1.2. Virulence Factors

There are multiple virulence factors contributing to its pathogenesis, including: (1) the surface structure lipopolysaccharides (LPS) and outer membrane proteins (OMP) facilitating adhesion to epithelial cells and efflux of harmful molecules; (2) flagellum and pili helping colonization in different niches; (3) secretion of exotoxins and proteases providing protection against immune surveillance and competing with other microorganisms, and (4) a quorum sensing (QS) system responding to external stressors and providing extraordinary flexibility [14,15]. These virulence factors contribute to acute infection and effective inhaled antibiotics are commonly applied to eradicate pathogens and prevent chronic infection.

#### 2.1.3. Adaptation and Persistence

Under the stress of declined lung function, and pulmonary inflammation, *P. aeruginosa* undergoes different genetic, morphological, and physiological changes for chronic adaptation (Figure 2). An impaired DNA repair system allows hypermutation of *P. aeruginosa*, generating multiple adaptive phenotypes for chronic adaptation [16]. Transition from acute to chronic *P. aeruginosa* phenotypes included different characteristics. Mucoid phenotype is a crucial morphological change characterized by overproduction of alginate, and the alginate-containing matrix can provide protection against phagocytosis and destruction by antibiotics [17,18]. Excessive alginate can promote biofilm formation, hindering bacteria penetration [19]. In biofilm, bacterial persister cells with low proliferation rates appear, which can survive longer there [20]. Small colony variants (SCV), also called auxotrophs, will also be developed and characterized by increased exopolysaccharide production and biofilm formation, also promoting *P. aeruginosa* persistence [21]. Virulence factors will be reduced or switched off, to avoid immune recognition and reduce energy expenditure, and the metabolic pathway will be adjusted to adapt anaerobic lung environment caused by excessive mucus [19]. Multi-drug resistant phenotypes can also be developed after intense drug therapy [22]. In conclusion, all those adaptive changes facilitate persistence and eventually cause failure of eradication and chronic pulmonary infection in CF patient [16].

### 2.2. Staphylococcus aureus

#### 2.2.1. Prevalence

*S. aureus* is the predominate gram-positive, spherically shaped bacterium colonized in CF children airways isolated of approximately 70% CF children. Due to intense penicillin treatments, methicillin-resistant *S. aureus* (MRSA) appears and has experienced a significant increase of about fivefold in recent years [23,24]. The recent two-year period encountered a slight decline in both MRSA and general *S. aureus* prevalence possibly due to effective early eradication protocols [3].

#### 2.2.2. Virulence Factors

The virulence factors of *S. aureus* include the secretion of multiple toxins such as Panton-Valentine leucocidin (PVL), which can induce necrosis of macrophages, neutrophils and monocytes, and cause severe infections [23,25]. They can also secrete multiple protein and non-protein factors, and immune evasion factors, which can facilitate host colonization and avoid immune surveillance [26].

#### 2.2.3. Adaptation and Persistence

Similar to *P. aeruginosa*, under the pressure of the immune system and antibiotics, both methicillin-sensitive *S. aureus* (MSSA) and MRSA can undergo several adaptative mechanisms for persistent infection, including, phenotypic transformation into metabolically inactive, slow-growing, small colony variants (SCVs), biofilm formation, anaerobic growth, and the emergence of persister cells (Figure 3) [23].

### 2.3. Other Bacteria

Other bacteria were also detected in CF airways, including *Nontuberculous Mycobacteria* (NTM)*, Stenotrophomonas maltophilia* and *Achromobacter*, all of which are associated with a higher risk of lung transplantation and increased mortality [25]. Anaerobes were also found in CF airways and enhanced *P. aeruginosa* colonization and virulence [1,25]. These bacteria have evolved different mechanisms of adaptation. For example, the most common NTM subtype, *Mycobacterium. avium* complex (MAC), can invade respiratory epithelial cells and phagocytic cells for persistent chronic infection [27].

## 3. Inhaled Antibiotics Treating Chronic *P. aeruginosa* Pulmonary Infection in Patients with Cystic Fibrosis

Systemic (intravenous and oral) antibiotics are available to treat chronic *P. aeruginosa* pulmonary infection in CF patients. Azithromycin, a second-generation macrolide interfering with bacterial protein synthesis by binding to 50S ribosomal subunit, has been demonstrated to be effective when improving lung function and reducing pulmonary exacerbations in CF patients chronically infected by *P. aeruginosa* [28]. However, the formation of biofilms significantly hinders drug penetration to persistent pathogens, so several times fold greater drug concentrations are required to reach the same effect as the pathogens without biofilm, which increases the risk of systemic toxicity significantly [29].

Compared to the systemic antimicrobial treatment, inhaled treatments lead to a more rapid onset of action, greater airway concentrations, and minimized systemic adverse effects, herein is now favored as a standard therapy for treating chronic *P. aeruginosa* pulmonary infection in CF patients [9,30]. Currently, there are two FDA-approved inhaled antibiotics: tobramycin and aztreonam. Interestingly, another two, levofloxacin and colistin, are only approved in the European Union and Canada and are still undergoing extensive phase III studies (Table 1; Figure 4) [29].

### 3.1. Tobramycin

#### 3.1.1. Mechanism of Action

Tobramycin is an aminoglycoside that binds to 30S ribosomal subunits and inhibits bacterial protein synthesis (Figure 4) [37].

#### 3.1.2. Efficacy

It is the most extensive study of inhaled antibiotics in targeting chronic *P. aeruginosa* infection in CF patients. Many clinical studies have confirmed the effectiveness of tobramycin inhalation solution (TIS) in both improving pulmonary function and reducing *P. aeruginosa* density and reducing the hospitalization rate (Table 2) [38,39,40,41]. A powder form of TIS (TIP) has also been investigated in clinical studies showing comparable efficacy in improving lung function and also an improved convenience due to shorter administration time [42,43].

#### 3.1.3. Safety

Clinical studies demonstrated adverse events after the TIS treatment, most of which were respiratory symptoms including coughing and pharyngitis, albeit the risk is not significantly different from the placebo. A good renal tolerability was also shown due to no significant change in creatinine levels [40]. The safety profile of TIP was generally comparable to TIS, except for the incidence of coughing, dysgeusia, and dysphonia due to increased irritation to the throat and airways caused by the powder itself [42,43].

#### 3.1.4. Indications

The first use of TIS was approved by the FDA in 1997, with the dosage at one ampoule (300 mg) per 5 mL solution twice daily for 28 days 28 day on/off cycle (Table 1) [58]. Another TIS formulation with the dosage of 300 mg/4 mL twice daily was approved in 2012 (Table 1) [59]. In 2013, the powder form of TIS (TIP) was approved with the recommended dose of four 28 mg capsules twice daily for a 28 day on/off cycle (Table 1) [60]. The doses of all three formations should be taken as close to 12 h apart and were recommended for patients aged 6 years and older due to the lack of studies for other populations. In 2015, a more concentrated, nebulized tobramycin, with a dose of 170 mg/1.7 mL was approved in Europe, which displayed a lower systemic exposure, but a comparable safety profile to TIS (Table 1) [61].

### 3.2. Aztreonam

#### 3.2.1. Mechanism of Action

Aztreonam is another FDA-approved inhaled antibiotic for chronic *P. aeruginosa* infection in CF patients (Table 1). It is a synthetic beta-lactam that disrupts bacterial cell wall synthesis by blocking peptidoglycan crosslinking (Figure 4) [62].

#### 3.2.2. Efficacy

Inhaled aztreonam has also been widely investigated and multiple clinical studies indicated that 75 mg three times a day improved lung function and reduced *P. aeruginosa* density significantly, and the safety profile was not inferior to the placebo (Table 2) [46,47,63]. When compared with the twice daily dosage at 300 mg/5 mL of TIS, the inhaled aztreonam demonstrated superiority over TIS in terms of improving pulmonary functions, reducing hospitalization, and the need for additional antibiotics [48].

#### 3.2.3. Safety

Reported adverse events included coughing, wheezing, pharyngolaryngeal pain, and more rarely, bronchispasm and rash (comparable to placebo) [46,47,63]. Aztreonam resulted in more drug-related adverse events than TIS, which is thought to be related to the bias from the unblinded study, and further investigations are yet to be conducted [48].

#### 3.2.4. Indications

Inhaled aztreonam, was approved by the FDA in 2010 at a dose of 75 mg/mL three times daily for 28 days [64]. It has only been approved for CF patients aged 7 years or older, and the safety and efficacy information for pregnant women and children is limited [64].

### 3.3. Levofloxacin

#### 3.3.1. Mechanism of Action

Levofloxacin is a fluoroquinolone antimicrobial which inhibits bacterial DNA replication by targeting bacterial-specific DNA gyrase and topoisomerase IV, causing failure of cell division and death (Figure 4) [29,65].

#### 3.3.2. Efficacy

Compared to tobramycin and aztreonam, levofloxacin has more rapid and complete bactericidal activity against mucoid and multidrug resistant *P. aeruginosa* isolates and it is also more potent against biofilm formation, retaining a minimally affected minimum inhibitory concentration (MIC) [66]. Three clinical randomized control trials have been reported for levofloxacin inhalation solution (LIS), including one dose-finding phase II trial and two-phase III trials (Table 2).

The phase II trial lasted 56 days (28 days treatment + 28 days follow-up) and included 151 CF subjects, which were randomized into 4 groups (120 mg/1.2 mL once daily, n = 38; 240 mg/2.4 mL once daily, n = 37; 240 mg/2.4 mL twice daily, n = 39; placebo, n = 37) [49]. Compared with the placebo, on day 28, the LIS treatment significantly reduced sputum *P. aeruginosa* density, among which the largest dosage of 240 mg/2.4 mL twice daily was associated with the greatest reduction of −0.73 log colony-forming units (CFU)/g sputum (placebo, +0.23 log cfu/g sputum; *p* = 0.001). The largest improvement in pulmonary function represented as a percent predicted FEV_1_ (ppFEV_1_%) relative change was also observed at 240 mg/2.4 mL in the twice daily group of + 8.6% (placebo, −2.4%; *p* = 0.0008). The risk of additional antibiotics requirement resulted in the largest reduction of 79% at 240 mg/2.4 mL in the twice daily group (*p* < 0.001), compared to 71% at 120 mg/1.2 mL everyday (*p* = 0.007) and 61% at 240 mg/2.4 mL everyday (*p* = 0.021). LIS was also demonstrated to be well-tolerated, and its resistance was not affected by biofilm formation. This study indicated that the 240 mg twice daily dosage of LIS has resulted in the best clinical and microbiological benefits with a good safety profile.

The dosage of 240 mg/2.4 mL twice daily was chosen for the following phase III trial due to its highest efficacy. The first randomized, but open-labelled phase III trial, was designed to compare the efficacy and safety of 240 mg/2.4 mL twice daily LIS (n = 189) to 300 mg/5 mL twice daily TIS (n = 93) in treating CF patients, which lasted 3 cycles, each of which contained 28 days of on treatment and 28 days of follow up off treatment [50]. On day 168, the LIS group resulted in a smaller proportion of hospitalization compared to the TIS group (17.5% vs. 28.0%; *p* = 0.04), and a longer median time was required for additional antibiotics (141 vs. 110 days; *p* = 0.04). The difference on day 28 was not statistically significant, in terms of (1) ppFEV1% change, with the least squares (LS) mean between-group difference (LIS minus TIS) of only 1.86% (95% CI, −0.66 to 4.39%) and (2) sputum *P. aeruginosa* density, with a LS mean difference of 0.44 log cfu/g (95% CI: −0.01 to 0.88), and those relationships are more complex. Generally, LIS showed no inferiority to TIS in treating CF chronic pulmonary infection.

In another double-blinded, placebo-controlled phase III trial, 220 CF subjects were randomized into 240 mg/2.4 mL twice daily LIS treatments with another 110 subjects randomized into a placebo group, lasting for a 56-day cycle (28 on + 28 off therapy) [67]. This trial did not reach its primary endpoint (i.e., time to an exacerbation of CF pulmonary disease) of demonstrating a LIS superiority over the placebo, which may be attributed to several reasons including unsimilar study populations and insufficient LIS concentration. Furthermore, LIS groups had a larger proportion of subjects with more than 3 exacerbations within 12 months prior to the trial (34.1% vs. 20%, *p* = 0.011), causing an imbalanced baseline for exacerbation hazard. On day 28, the LIS group manifested greater relative change in ppFEV_1_% (LS mean difference: 2.42%; 95% CI 0.53 to 4.30%) and a greater reduction in sputum *P. aeruginosa* density (LS mean difference: −0.63 log_10_ CFU/g; 95% CI −0.95 to −0.30). Both differences between treatment groups were significant favoring the LIS group. No significant difference was shown in the median time for additional antibiotics and hospitalization between both groups (LIS, 59 days vs. placebo, 58 days).

Both phase III trials indicated at least no inferiority or a slight superiority of inhaled levofloxacin over tobramycin in terms of clinical and microbiologically benefits in treating chronic *P. aeruginosa* infections in CF patients. 

#### 3.3.3. Safety

Despite generally similar safety and tolerability profile, LIS has still been associated with increased risk of several adverse effects. According to LIS, dysgeusia had the most common adverse effect reported in 35.2% of the LIS groups in the placebo-controlled study (0% in placebo group) and 25.3% of the LIS group in LIS versus TIS study (0% in TIS group) [50,67,68]. In the LIS versus TIS study, 88 subjects were involved in an extension study for three additional LIS treatment cycles and 13.6% developed dysgeusia [69]. Coughing is another adverse effect showing higher incidences in the LIS group [50,67]. Adverse effects related to systemic administration including tendinopathies and neuropathies were not common in those clinical studies, indicating an advantage favoring the inhaled formulation [29,50,67,68].

#### 3.3.4. Indication

In 2015, LIS, was approved by the European Medicines Agency (EMA) and Canada to treat chronic *P. aeruginosa* infection in cystic fibrosis, in consideration of the benefits overweighting the risks [70,71]. The recommended dose is one ampoule (240 mg) twice a day, inhaled using exclusively a nebulizer system converting an ampoule into a fine mist (Table 1) [70]. The dose should be close to 12 h apart, but no less than 8 h apart. One treatment cycle is composed of 28 days on treatment and 28 days of follow-up off treatment, which can be repeated until obtaining optimal clinical results [70]. Its efficacy and safety profile have not been established in children under 18 years and pregnant women, so LIS should not be prescribed for those populations [72]. Due to its reported rare but severe adverse effects including aorta ruptures, despite a successful phase III trial in the USA, it has still not been approved by the FDA and further research on the safety profile and pharmacokinetics is required [73].

### 3.4. Colistin

#### 3.4.1. Mechanism of Action

Colistin disrupts the bacterial plasma membrane by interacting with LPS in the outer membrane of the gram-negative bacteria and displacing LPS-stabilizing ions Mg^2+^ and Ca^2+^, which causes derangement of the outer membrane, increases permeability of the cell envelope, and eventually, cell death (Figure 4) [74,75,76]. Colistimethate is a pro-drug, which is converted to active colistin after absorption. Colistimethate sodium, one available form of colistin with less toxicity, is preferred for administration by inhalation [77].

#### 3.4.2. Efficacy

As one of the earliest antibiotics used against *P. aeruginosa,* displaying high efficacy even against multi-drug resistant strains, inhalation therapy of colistin was introduced and investigated in early 1980 against CF chronic respiratory infection, and several clinical studies were conducted in the following decades (Table 2) [77].

In the first double-blinded, placebo-controlled clinical trial, 20 CF subjects received 1 million international units (MIU)/3 mL of colistin (colistin sodium methansulphonate) twice daily for 90 days, compared with another 20 CF subjects in the placebo group, and 2-week intravenous treatment with tobramycin plus beta-lactam was administered prior to the study [52]. The treatment group resulted in lower clinical scores (colistin; 1.8 vs. placebo; 4.7, *p* < 0.01, a higher score indicates more signs of pulmonary exacerbations), smaller mean FEV_1_% reduction (colistin; −11% vs. placebo; −17%, *p* < 0.05) and decreased inflammatory parameters, suggesting the superiority of colistin over the placebo in treating chronic infection. Colistin treatment was also well tolerated without the emergence of resistance. However, this colistin dose did not stop pulmonary function decline and large randomized, placebo-controlled trials were missing.

Hodson et al. conducted the first randomized trial comparing 300 mg/5 mL TIS (n = 52) with 80 mg (1 MIU)/3 mL of nebulized colistimethate sodium solution (n = 62) twice daily for 4 weeks [53]. The TIS group resulted in a significant increase of 6.7% in FEV1% and predicted a change from the baseline (*p* = 0.006), while the colistin group caused an insignificant rise of only 0.37% (*p* = 0.473). Both groups reduced *P. aeruginosa* density by 4 weeks, whereas the changes were both not significant, but slightly favoring the TIS treatment (TIS, −0.86 log_10_ CFU/g, *p* < 0.001; Colistin, −0.6 log_10_ CFU/g, *p* = 0.007). Patients in both groups had received different amounts of nebulized/intravenous colistin or tobramycin prior to the trial, possibly providing bias to the results, but generally inhaled tobramycin is superior to colistin in terms of improving lung function in CF patients with chronic pulmonary infection. Both treatments were similarly tolerated. A follow-up study extended the current treatment for a further 5 months and the results indicated an increasing the slope to 0.35% per month regarding FEV_1_% predicted a change over time in the TIS group, compared to the −0.88% decreasing slope in the colistin group (*p* = 0.0002), which further confirms the superiority of TIS over nebulized colistin on reversing pulmonary decline [78].

More recently, in another randomized, open-labelled, phase III study, 187 CF subjects were randomized into a continually 24-week colistimethate sodium dry powder for inhalation (CDPI) (one capsule of 1.6625 MIU, twice daily) treatment group, compared to 193 CF subjects in the TIS (300 mg/5 mL, twice daily) group (3 cycles with each containing 28 on/off treatment) [54]. In week 24, the mean changes in FEV_1_% predicted were 0.964% for the CDPI group and 0.986% for TIS (adjusted mean difference, −0.98%; 95% CI, 2.75% to 0.86%), indicating no inferiority of CDPI over TIS in terms of improving pulmonary functions. The comparable results contradict the previous trial suggesting inferiority of colistin over tobramycin, which may be attributed to the reason that a higher dosage of colistin was used in dry powder and all patients were treated with at least 2 cycles of inhaled tobramycin before the trial [77]. Patients favor colistin regarding treatment burden maybe because of less time required for inhalation (1 min vs. 15~20 min) [77].

Even though inhaled colistin did not show superiority over tobramycin in terms of improving lung function, it has the advantage of a slower resistance development [79,80]. In 1987, inhaled colistin therapy was chosen in the Danish CF center due to the emergence of tobramycin-resistant strains, and only 14 out of 120 patients have developed colistin-resistant strains since then, indicating a much slower development of resistance to colistin than tobramycin [81,82,83,84]. The rare occurrence of colistin resistance may be attributed to the fact that colistin can self-promote its cell envelope penetration and disrupt cell membrane irreversibly, preventing *P. aeruginosa* from modifying LPS [77]. Colistin has other advantages including a narrow spectrum, protecting normal bacteria in throat and gut from being disrupted, and functioning as an anti-endotoxin reagent [81].

#### 3.4.3. Safety

Nebulized colistin therapy can avoid systemic neuro- and nephrotoxicity and most adverse effects occur in the airways. Chest tightness and bronchospasm have been reported in CF patients caused by inhaled colistin [85,86]. Compared with the TIS, colistimethate sodium solution for inhalation resulted in similar side effects including coughing, dyspnea and pharyngitis, but increased coughing was more prevalent in the colistin-treatment group. The more recent CDPI formulation causes a higher incidence of coughing (75.4% vs. 43.5%), taste distortion (62.6% vs. 27.5%) and throat irritation (45.5% vs. 28.0%), due to deposition in the pharynx, but improving inhalation techniques can reduce adverse effects [42,54]. A fatal case of acute respiratory distress syndrome (ARDS) was reported in a 29-year-old woman with CF after treatment with 75 mg of nebulized colistimethate sodium twice daily, which is thought to be due to excessive conversion of pro-drug colistimethate sodium to the less-tolerable biologically active colistin form [63]. A recent long-term observational study including 4969 CF patients treated with CDPI confirmed that its safety profiles were similar to other inhaled antibiotics [87].

#### 3.4.4. Indications

Colistimethate sodium inhalation powder was approved by the EMA in 2012 to treat chronic *P. aeruginosa* lung infections in cystic fibrosis, and the recommended dose for children was 1–2 MIU two to three times per day greater than 2 years (0.5–1 MIU twice daily for children younger than 2), inhaled using suitable nebulizers (Table 1) [88]. Due to the lack of safety data, pregnant and lactating women should only use it when the benefits overweigh the potential risks. Possible sides effects include skin rash, drug fever, sore throat or mouth, and more severely, nephron- and neurotoxicity [88]. Due to the limited detailed pharmacokinetic data and unstandardized dosing, inhaled colistin has not been FDA-approved in the USA, and further phase III and toxicity studies are planned to confirm its efficacy and safety in CF patients [89,90].

### 3.5. Amikacin Liposome Inhalation Suspension

#### 3.5.1. Mechanism of Action

Amikacin is a broad-spectrum aminoglycoside, inhibiting bacterial protein synthesis by binding to the 30S ribosomal subunit (Figure 4) [91]. Its liposomal formulation and amikacin liposome inhalation suspension (ALIS), was designed for inhalation, facilitating penetration into biofilm, and sputum [56].

#### 3.5.2. Efficacy

Due to the high efficacy and good tolerability, ALIS has been widely used against MAC lung disease, while its ability to treat chronic *P. aeruginosa* infection in CF is also currently being studied (Table 2).

In a double-blinded phase II study, 105 CF subjects were randomized into 4 once-daily ALIS groups (70 mg, n = 7; 140 mg, n = 5; 280 mg, n = 21; 560 mg, n = 36) and 1 placebo group (n = 36) for a 28 day on/off cycle [55]. The 560 mg dose group resulted in the highest and most sustained increase in FEV_1_ compared to the placebo (0.093 L ± 0.203 vs. −0.032 L ± 0.119; *p* = 0.003) on day 56, as well as the largest reduction in *P. aeruginosa* sputum density on day 35. In this study, 560 mg dosage of ALIS was shown to be the most effective in treating CF *P. aerugonisa* infection. The efficacy of 560 mg once daily of ALIS was further supported by two other phase II studies [92].

In another open-labelled phase III study, 590 mg/8.4 mL of ALIS once daily (n = 152) was compared with 300 mg/5 mL of TIS twice daily (n = 150), for three 28 day on/off treatment cycles [56]. Regarding relative change in FEV_1_%, on day 168, the LS mean difference (ALIS-TIS) was −1.31% (95% CI: −4.95–2.34), suggesting noninferiority of ALIS over TIS to improve lung function. Mean reductions in *P. aeruginosa* sputum density were fluctuated during cycles but reduced at the end in both groups. With similar safety profiles shown, this study suggested a comparability of ALIS over TIS in treating chronic CF *P. aerugonisa* infection. In a following extended study, ALIS treatment was extended for another 12 cycles (96 weeks) with 206 participants from the previous phase III study [93]. At the end of the study, both lung function and *P. aerugonisa* sputum density remained stable, suggesting the efficacy of ALIS over the 2-year treatment period.

#### 3.5.3. Safety

In the placebo-controlled study, the adverse events profile was similar between different treatment and the placebo groups, with the most adverse events as respiratory symptoms [55]. One major difference is dysphonia, which was only reported in the high-dose 560 mg ALIS group (8%). Compared with TIS, ALIS treatment led to more drug-related treatment emergent adverse events (TEAEs) including dysphonia (10.8% vs. 2.7%), coughing (8.8% vs. 2.1%), and pulmonary exacerbations (6.8% vs. 2.1%) [56].

#### 3.5.4. Indications

ALIS was approved for treating lung disease caused by NTM in 2018, with a recommended dose of 590 mg/8.4 mL once daily by oral inhalation (Table 1) [94]. However, due to the lack of clinical benefit data, this drug was only limited to patients who do not have CF [95]. The research on its appropriate dosage and impact on persistent *P. aeruoginosa* in CF patients is limited and further phase III studies are required. ALIS also appears beneficial in terms of treating MAC lung disease in CF patients in one French study, but accessible data is extremely limited and further investigation is required for approval [96].

### 3.6. Murepavadin

#### 3.6.1. Mechanism of Action

Murepavadin is an antibiotic that exerts a novel mechanism of action by binding to an Gram-negative outer membrane protein, lipopolysaccharide transport protein D (LptD), inhibiting its LPS transport function, disrupting outer membrane assembly and eventually causing cell death (Figure 4) [97].

#### 3.6.2. Efficacy

Preclinical and clinical studies have shown a promising efficacy of murepavadin against multi-drug resistant (MDR) *P. aeruginosa* [97]. However, due to significant renal toxicity, intravenous murepavadin to treat pneumonia has been withdrawn in phase III trials; instead, inhaled murepavadin with less systemic exposure and limited toxicity is supported by the Bernardini group, who investigated potent activity against *P. aeruginosa* and the good tolerability after aerosol administration [98,99]. Ekkelenkamp and colleagues found that murepavadin has a lower MIC_50_ of 0.12 mg/L (aztreonam 8 mg/L; tobramycin 1 mg/L; colistin 1 mg/L) and MIC_90_ of 2 mg/L (aztreonam 128 mg/L; tobramycin 16 mg/L; colistin 2 mg/L), suggesting more potent antimicrobial activity against *P. aerugonisa* isolated from CF patients than other antibiotics [100]. Moreover, it showed high activity against biofilms of CF, *P. aeruginosa*, and its antimicrobial activity remained unchanged in the lung surfactant and artificial sputum, which further supports its for inhalation in pulmonary infections [98,101].

Preclinical studies of inhaled murepavadin in treating CF chronic respiratory infection have been completed recently, indicating promising efficacy and a favorable safety profile compared to the intravenous formulation [57]. A phase I study evaluating its safety and pharmacokinetics in healthy adults was initiated in late 2020 with results expected in 2022, and a following phase 1b/2a study conducted in CF adults is also planned to be started in 2023 (Table 2) [57]. Further phase I and II studies are required to determine the dosage.

#### 3.6.3. Safety

Regarding inhaled murepavadin, only preclinical studies were currently finished, which demonstrated an encouraging safety profile, whereas further phase I and II studies are required to assess its safety in both healthy people and CF patients [57].

#### 3.6.4. Indications

Optimal dosage is yet to be determined in future phase I and II studies.

## 4. Inhaled Antibiotics Treating Chronic *S. aureus* Pulmonary Infection in CF Patients

Before the emergence of MRSA in the 1990s, oral rifampicin and fusidic acid were used commonly to treat chronic *S. aureus* infections in CF patients [102]. The increasing risk of MRSA complicated the problem and novel inhaled antibiotics alone or in combination with conventional oral ones are under investigation on the impact of chronic MRSA infections in CF patients. 

### 4.1. Vancomycin

Available inhaled antibiotic strategies targeting chronic *S. aureus* infections in CF patients are significantly limited. Inhaled vancomycin, a glycopeptide antibiotic, has been studied recently. It acts by inhibiting cell wall synthesis in Gram-positive bacteria and possesses a potent efficacy against *S. aureus*, especially MRSA (Figure 5) [103].

Although inhaled vancomycin alone or with other oral antibiotics showed efficacy on eradiating newly acquired MRSA in several studies, its impact on chronic *S. aureus* infection has rarely been investigated [106,107,108,109]. An early case report demonstrated reduction of MRSA density but failure of MRSA eradication after aerosolized vancomycin treatment to a 10-year CF patient chronically infected with MRSA [109]. In another randomized, double-blinded study, 29 CF patients with chronic MRSA infection were treated with multiple oral antibiotics for 28 days, 14 of which also received an additional dose of inhaled vancomycin [110]. Results were analyzed at 3 time points (end of, one month after, three months after the treatment). However, despite absolute MRSA sputum density reduction in the inhaled vancomycin group, it is not significantly different from the inhaled placebo group at any time point, and both reached low MRSA eradication rates of approximately only 20% one month and three months after treatment. Inhaled vancomycin has not been shown to lead to greater pulmonary improvements at any time point and may be also related to the risk of bronchospasm. Confounded by the small sample size, no definitive impacts of inhaled vancomycin on treating chronic *S. aureus* infection in CF patients can be concluded.

### 4.2. Fosfomycin

Fosfomycin is a novel phosphonic acid antibiotic disrupting both Gram-positive and Gram-negative bacterial cell wall synthesis, which has a potent antimicrobial activity against *S. aureus* (both MRSA and MSSA) (Figure 5) [111]. A 4:1 combination of fosfomycin and tobramycin for inhalation (FTI) (160/40 mg or 80/20 mg) has been investigated to treat CF patients co-infected with *P. aeruginosa* and *S. aureus* [112]. Both sputum MRSA and MSSA density were reduced following both FTI dosage treatments and 11 out of 46 patients receiving FTI reached *S. aureus* eradication 28 day after treatment, which suggests a potential efficacy of FTI in treating *S. aureus* infection. Nevertheless, *S. aureus* results were confounded by whether *S. aureus* infections were newly acquired or chronic since the presence of *S. aureus* were not required for enrollment of the participants. The impact of FTI on chronic *S. aureus* infection requires further research.

In conclusion, there are very few studies collected about inhaled antibiotics targeting chronic *S. aureus* infections in CF patients, and significant microbiological and clinical benefits have not been adequately demonstrated, so further research is still required.

## 5. Selecting the Optimal Inhaled Antibiotic Therapy

Several factors should be considered when selecting inhaled antibiotics for chronic infections in CF patients. Firstly, and primarily, the microbiological and clinical effectiveness (antibiotic sensitivity) should be considered, including changes in sputum bacterial density, the lowest antibiotic concentration to inhibit 50% (MIC_50_) and 90% of isolates (MIC_90_), ppFEV_1_%, risk of pulmonary exacerbations, and hospitalizations. Secondly, adverse effects need to be evaluated. Thirdly, drug resistance should be compared and antibiotics with slower resistance development rates may be preferred due to prolonged treatment efficacy. Optimizing clinical outcomes, minimizing toxicity, and minimizing antimicrobial resistance are three guideline principles of antimicrobial stewardship in CF [113]. Moreover, the treatment burden will also affect patients’ choices. Patients will prefer drugs requiring less inhalation time and frequency, more convenient equipment, and less cost, but the consideration of treatment burden should not overweigh that of the medical needs [114]. With so many affecting factors, the optimal treatment is usually difficult to be determined. For instance, although colistimethate sodium dry powder has the advantages of a slower resistance rate and shorter inhalation time compared to other nebulizers (1 min vs. 5–20 min), its dry powder form may also lead to more respiratory discomfort and irritation, so its feasibility may be variable among patients (Table 1) [77].

Due to *P. aeruginosa* persistence, long term maintenance therapy of inhaled antibiotics is recommended, and patterns of cycling should be considered for optimization. 28-day on/off alternating monotherapy has been conventionally applied to most inhaled antibiotics due to the rationale that “drug holidays” allow populations to regain susceptibility (Table 2) [39,115]. In recent years, there was also a growing trend moving from monotherapy to a combination of antibiotics with different mechanisms of actions, to obtain better effectiveness [29,116].

### Combination Therapy

Despite considerable effectiveness, some CF patients may be intolerant or resistant to a certain inhaled antibiotic, causing failure of monotherapy. With the development of novel effective inhaled antibiotics, there is a growing interest in the continuous alternating inhaled antibiotic therapy (CAIT), in which multiple antibiotics are rotated monthly [29,117]. According to Dasenbrook and colleagues, after the approval of inhaled aztreonam, CF patients in the US Cystic Fibrosis Foundation Patient Registry (CFFPR) receiving more than 1 inhaled antibiotics annually more than doubled between 2009 and 2012 [118]. The introduction of additional inhaled antibiotics is predicted to provide many benefits. Primarily, inhaled antibiotics with different mechanisms of actions can generate synergistic effects and limit resistance. CF patients intolerant to one antibiotic can benefit from another class. For instance, double or triple combinations including colistin, an anti-pseudomonal beta-lactam, and aminoglycoside or rifampin, have been demonstrated to how additive the effects against *Pseudomonas* spp. [119]. Colistin, which can enhance the membrane penetration of other antibiotics, is often considered as an ideal combination antibiotic [119]. Moreover, the effective life of antibiotics can be extended with rotation of different antibiotic classes [118]. CAIT combining colistin, tobramycin and aztreonam was associated with improved FEV_1_ in one retrospective study [117]. Another double-blinded trial comparing continuous alternating inhaled aztreonam and tobramycin combination therapy with the conventional alternating inhaled tobramycin monotherapy was conducted in 2015 [51]. Although the differences in pulmonary function improvement and *P. aeruginosa* density reduction were not significant, a better reduction in exacerbation and hospitalization rates in the alternating combination therapy group suggested its potential superiority over monotherapy against chronic pseudomonal infection in CF. However, this study was hugely underpowered by limited study enrolment and a small sample size. Generally, regarding chronic *P. aeruginosa* infection in CF, a definitive advantage of combination therapy over intermittent monotherapy are significantly lacking and further evaluation is required. Regarding chronic and combination therapies for other bacteria including *S. aureus,* accessible evidence is further limited [114].

## 6. Conclusions and Future Perspectives

Under the pressure of the immune system and the antibiotic therapies, pathogens undergo different adaptative changes for persistence, resulting in chronic lung infections, and *P. aeruginosa* and *S. aureus* are the two most commonly colonized types in CF patients. Persistent bacteria are difficult to eradicate and inhaled antibiotic strategies with higher airway concentration with minimized systemic toxicity are urgently required to prevent progressive lung damage. Currently approved inhaled antibiotics on the market are hugely limited and most are still in the pipelines. Regarding persistent *P. aeruginosa,* only inhaled tobramycin and aztrenom are fully approved by the FDA, while the other two, inhaled levofloxacin and colistin, although showing microbiological and clinical benefits in the phase III trial, have not been FDA-approved due to adverse effects and limited pharmacokinetic details. Phase III trials of ALIS are still ongoing and encouraging results have already been shown. A Phase I study of murepavadin has been initiated. Available inhaled antibiotic protocols targeting chronic *S. aureus* infections are very few. Both inhaled vancomycin and FTI displayed possible efficacy, but results are hugely confounded. Selection of optimal inhaled antibiotics is based on multiple factors and a gold standard is hard to be determined.

Future research comparing clinical and microbiological results as well as safety and treatment burden of different inhaled antibiotics are consistently required to navigate the treatment of chronic pulmonary infections in CF. Moreover, the superiority of combination therapy over monotherapy should be further assessed in large-size studies.

## Figures and Tables

**Figure 1 antibiotics-12-00484-f001:**
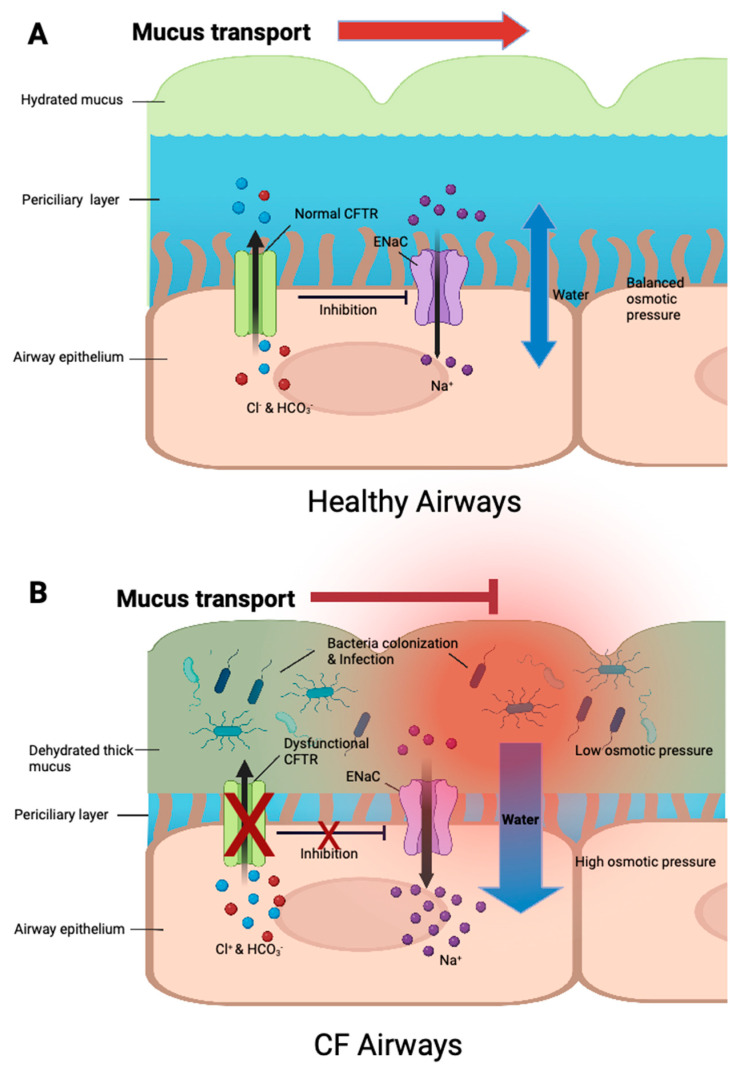
The regulation of ion transport and mucus hydration in healthy and CF airway surface. (**A**) In healthy airways, functional CFTR regulate Cl^−^ and HCO_3_^−^ secretions, and inhibits epithelial Na^+^ channels (ENaC), contributing to the balance of intracellular and extracellular ion concentrations as well as osmotic pressure. Water movement is balanced, and mucus is well-hydrated, allowing the removal of pathogens. (**B**) In CF airways, dysfunctional or absent CFTR prevents Cl^−^ and HCO_3_^−^ secretions, resulting in intracellular ion accumulation. ENaC activity cannot be inhibited, causing excessive Na^+^ absorption, which further contributes to ion accumulation. Excessive water is drawn inside due to high intracellular osmotic pressure, leading to mucus dehydration, failure of ciliary clearance, pathogen colonization and eventually pulmonary infections.

**Figure 2 antibiotics-12-00484-f002:**
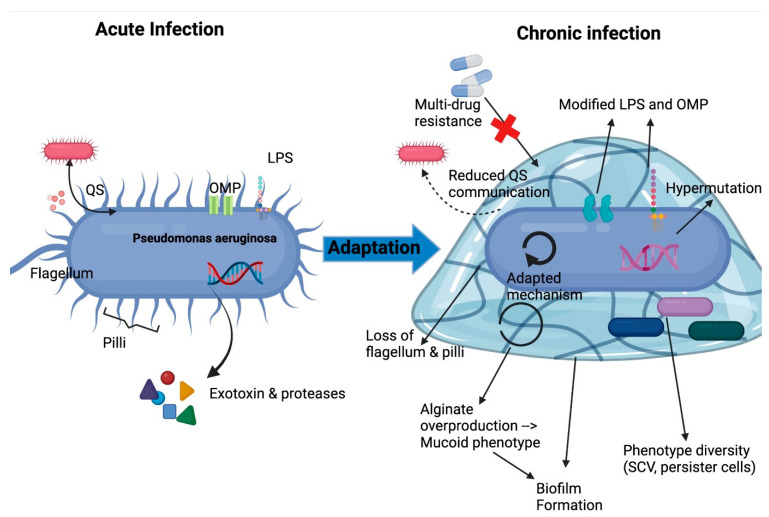
Adaptation mechanisms of *P. aeruginosa* in lung environment leading to chronic infection. LPS, lipopolysaccharide; OMP, outer membrane protein; QS, quorum sensing; SCV, small colony variants. Adapted from Jurado-Martin et al. [14].

**Figure 3 antibiotics-12-00484-f003:**
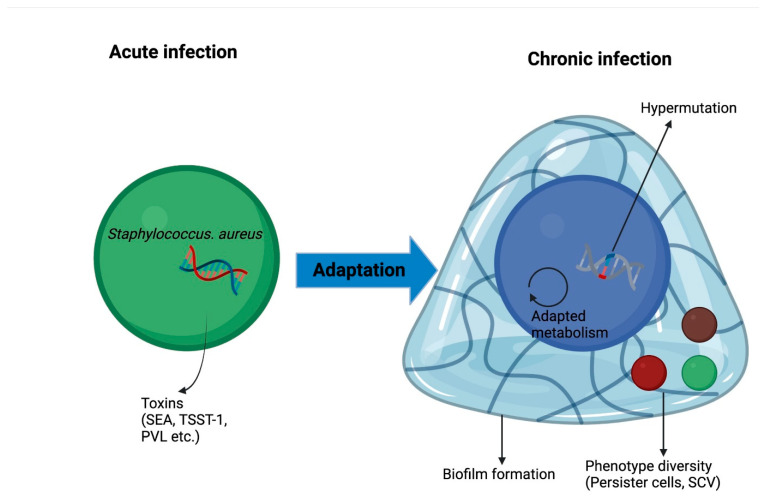
Adaptation mechanisms of *S. aureus* in lung environment leading to chronic infection. PVL, Panton-Valentine leucocidin; SEA, staphylococcal enterotoxin A; SCV, small colony variants; TSST-1, toxic syndrome toxin.

**Figure 4 antibiotics-12-00484-f004:**
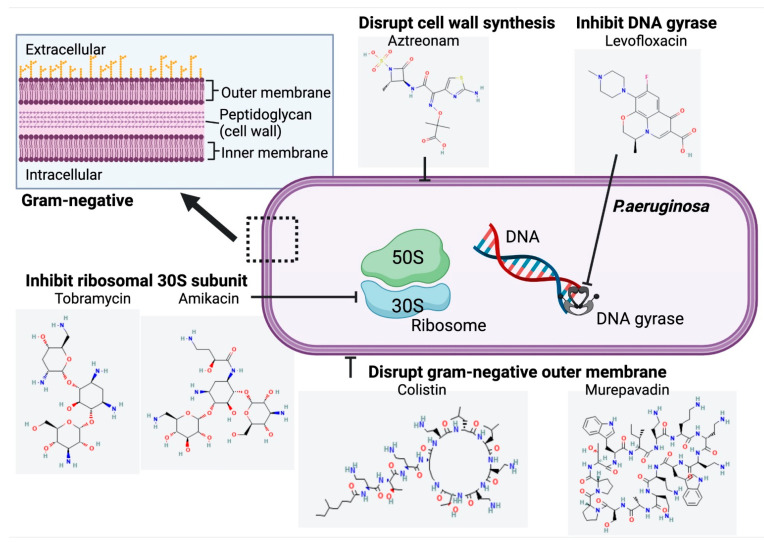
Structure and Mechanism of Action of Inhaled Antibiotics Targeting Chronic *P. aeruginosa* Infection in CF. Chemical structures accessed from PubChem [31,32,33,34,35,36].

**Figure 5 antibiotics-12-00484-f005:**
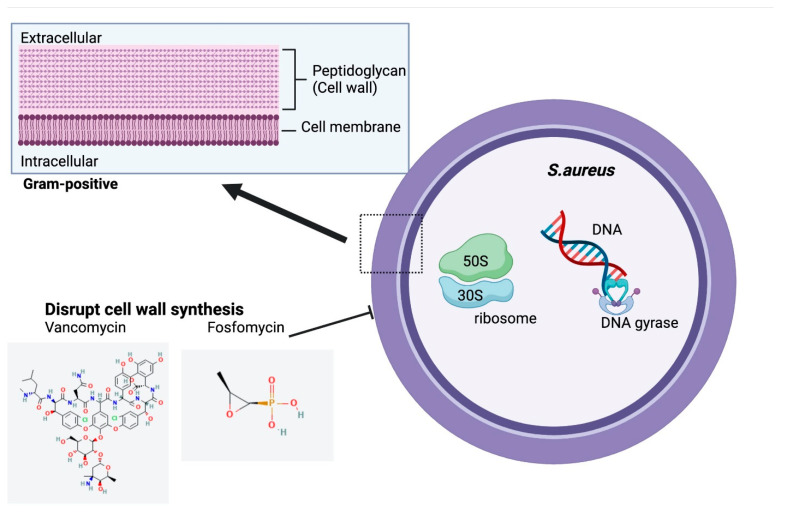
Structure and Mechanism of Action of Inhaled Antibiotics Targeting Chronic *S.aurues* Infection in CF. Chemical structures accessed from PubChem [104,105].

**Table 1 antibiotics-12-00484-t001:** Summary of Inhaled Antibiotics to Treat Chronic *P. aeruginosa* Pulmonary Infections in CF Patients.

	Pharmaceutical Form	Dose	Dose Frequency	Inhalation Time	FDA Status
Tobramycin	Clear slightly yellow nebulizer solution	300 mg/5 mL	Twice daily	~15 min	FDA approved
	Clear yellowish nebulizer solution	300 mg/4 mL	Twice daily	~15 min	FDA approved
	Dry powder in capsule	4 × 28 mg (4 capsules) = 112 mg	Twice daily	~5 min	FDA approved
	Clear, slightly yellow solution	170 mg/1.7 mL	Twice daily	~4 min	Approved in Europe
Aztreonam	White powder diluted in diluent for nebulizer solution	75 mg/mL	Three times daily	~2–3 min	FDA approved
Levofloxacin	Clear pale yellow nebulizer solution	240 mg/3 mL	Twice daily	~5 min	Phase III
Colistin	Sterile white dry power	1–2 MIU (80–160 mg)	Two to three times daily	~1 min	Phase III
Amikacin liposome inhalation suspension	Milky white nebulizer dispersion	590 mg/8.4 mL	Once daily	NA	Phase III
Murepavadin	NA	NA	NA	NA	Phase I

MIU, million international unit; NA, not available.

**Table 2 antibiotics-12-00484-t002:** Randomized studies on inhaled antibiotics in treating *P. aerugonisa* chronic pulmonary infection in CF patients.

	Reference	Year	Type	Treatment	Duration (Days)
Tobramycin	Ramsey et al. [39]	1999	Double-blinded	TIS 300 mg/5 mL BID, n = 258; Placebo, n = 262	168 = (28 on + 28 follow-up) × 3
	Chuchalin et al. [40]	2007	Double-blinded, multicenter	TIS 300 mg/4 mL BID, n = 161; Placebo, n = 86	168 = (28 on + 28 follow-up) × 3
	Lenoir et al. [41]	2007	Double-blinded	TIS 300 mg/4 mL BID, n = 29; Placebo, n = 30	56 = 28 on + 28 follow-up
	Nasr et al. [38]	2010	Double-blinded, multicenter	TIS 300 mg/5 mL BID, n = 16; Placebo, n = 16	168 = (28 on + 28 follow-up) × 3
	Konstan et al. (EVOLVE study) [43]	2011	Double-blinded, multicenter	TIP 112 mg BID n = 46; Placebo, n = 49	168 = (28 on + 28 follow-up) × 3
	Konstan et al. (EAGER study) [42]	2011	Open-labelled, Multicenter	TIP 112 mg BID, n = 308; TIS 300 mg/5 mL BID, n = 209	168 = (28 on + 28 follow-up) × 3
	Quittner et al. [44]	2012	Double-blinded	TIP 112 mg BID, n = 32; Placebo, n = 30	56 = 28 on + 28 follow-up
Aztrenom	McCoy et al. [45]	2008	Double-blinded, multicenter	AZLI 75 mg BID, n = 69; AZLI 75 mg TID, n = 66; Placebo, n = 76	28 + 36 follow-up
	Retsch-Bogart et al. [46]	2009	Double-blinded, multicenter	AZLI 75 mg TID, n = 80; Placebo, n = 84	28
	Wainwright et al. [47]	2011	Double-blinded, multicenter	AZLI 75 mg, TID, n = 76; Placebo, n = 81	28 + 14 follow-up
	Assael et al. [48]	2013	Open-labelled, multicenter	AZLI 75 mg TID, n = 136; TIS 300 mg BID, n = 132	168 = (28 on + 28 follow-up) × 3
Levofloxacin	Geller et al. [49]	2014	Phase II, double-blinded	LIS 120 mg/1.2 mL QD, n = 38; LIS 240 mg/2.4 mL QD, n = 37; LIS 240 mg/2.4 mL BID, n = 39; Placebo, n = 37	56 = 28 on + 28 follow-up
	Stuart Elborn et al. [50]	2015	Phase III, open-labelled	LIS 240 mg/2.4 mL BID, n = 189; TIS 300 mg/5 mL BID, n = 93	168 = (28 on + 28 follow-up) × 3
	Flume et al. [51]	2016	Phase III, double-blinded	LIS 240 mg/2.4 mL BID, n = 220; Placebo, n = 110	56 = 28 on + 28 follow-up
Colistin	Jensen et al. [52]	1987	Double-blinded	COL 1 MIU/3 mL BID, n = 20; Placebo, n = 20	90
	Hodsen et al. [53]	2001	Multicenter	COL 1 MIU/3 mL BID, n = 62; TIS 300 mg/5 mL BID, n = 52	28
	Schuster et al. [54]	2013	Phase III, open-labelled, multicentfoer	COL-P 1.6625 MIU BID, n = 187; TIS 300 mg/5 mL BID, n = 193	168 = (28 on + 28 follow-up) × 3
Amikacin Liposome Inhalation Suspension	Clancy et al. [55]	2013	Phase II	ALIS 70 mg OD, n = 7; ALIS 140 mg OD, n = 5; ALIS 280 mg OD, n = 21; ALIS 560 mg OD, n = 36; Placebo, n = 36	168 = (28 on + 28 follow-up) × 3
	Bilton et al. [56]	2020	Phase III	ALIS 590 mg/8.5 mL OD, n = 152; TIS 300 mg/5 mL BID, n = 150	168 = (28 on + 28 follow-up) × 3
Murepavadin	Spexis [57]	2021	Phase I	NA	NA

TIS, tobramycin inhalation solution; TIP, tobramycin inhalation dry powder; AZLI, aztreonam for inhalation solution; LIS, levofloxacin inhalation solution; COL, colistimethate sodium inhalation solution; COL-P, colistimethate sodium dry powder; QD, once a day; BID, twice a day. TID, three times a day; MIU, million international unit; NA, not available.

## Data Availability

The data that support the findings of this study are available from the corresponding author upon reasonable request. Some data may not be made available because of privacy or ethical restrictions.

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
