# Peer review of "Current and Emerging Inhaled Antibiotics for Chronic Pulmonary Pseudomonas aeruginosa and Staphylococcus aureus Infections in Cystic Fibrosis"

_antibiotics, 2023, doi:10.3390/antibiotics12030484_

Round 1
Reviewer 1 Report
I found this review to be concise and focused. The authors do an excellent job reviewing the most commonly used inhaled antibiotics for CF, including their modes of action, their effectiveness in treating different bacterial strains and pulmonary function, and any adverse effects associated with their use. The figures and tables are well-designed and aid the reader in comprehending the large body of information. I find no flaws in the organization or in the referencing of previous studies. However, there are grammar and spelling corrections throughout the manuscript that need to be corrected. In addition, there are many abbreviations used throughout the study. It would be very helpful to have a table of abbreviations to which a reader can refer. I have attached the manuscript where I have marked where grammatical/spelling/working changes are needed. A "?" next to a section indicates that I don't understand what the the authors are saying and the wording should be altered for clarity.

Author Response
Reviewer 1:
I found this review to be concise and focused. The authors do an excellent job reviewing the most commonly used inhaled antibiotics for CF, including their modes of action, their effectiveness in treating different bacterial strains and pulmonary function, and any adverse effects associated with their use. The figures and tables are well-designed and aid the reader in comprehending the large body of information. I find no flaws in the organization or in the referencing of previous studies. However, there are grammar and spelling corrections throughout the manuscript that need to be corrected. In addition, there are many abbreviations used throughout the study. It would be very helpful to have a table of abbreviations to which a reader can refer. I have attached the manuscript where I have marked where grammatical/spelling/working changes are needed. A "?" next to a section indicates that I don't understand what the the authors are saying and the wording should be altered for clarity.
Authors’ response: We appreciate the editor’s suggestion and amended the grammar and spelling errors in the manuscript (multiple sections), and added an abbreviation table at the start (p.1, line 24ff).
Reviewer 2 Report
I found this review article interesting for the readers Antibiotics and followed the journal Antibiotics’ scope. I would recommend the review article be published in Antibiotics after minor corrections.
I would recommend the article be published in Antibiotics after minor corrections.
The author needs to address the following comments/corrections.
1. The author could have discussed about Cystic Fibrosis, and its symptoms with available therapies.
2. The author could have shown a diagram of the disturbance of secretion of Cl- and HCO3 (epithelial Na+ channel (ENaC) dysfunction) by defective CFTR resulting mucus dehydration.
3. Follow the same pattern of discussion for S. aureus as in case of P. aeruginosa.
3.1.1. Prevalence
3.1.2. Virulence factors
3.1.3. Adaptation & Persistence
4. The author could talk about different types of nebulizers in the introduction.
5. Space missing for “300mg/5mL” (correct all).
6. The author could discuss a bit the mechanism of action of inhaled antibiotics targeting chronic P.aeruginosa and S. aureus infection.
7. For the section 3, follow same pattern (Mechanism of Action, Efficacy, Safety, and Indication) for all antibiotics.
8. Remove underline from sections 3.4.1, 3.4.2, 4.1 and 4.2 (335-346, 419-452).
9. Provide data in the table for section 5 and 6.
10. The author should correct the format of references wherever needed (e.g Year Bold, Volume Italic etc).
Author Response
Reviewer 2:
I found this review article interesting for the readers Antibiotics and followed the journal Antibiotics’ scope. I would recommend the review article be published in Antibiotics after minor corrections.
I would recommend the article be published in Antibiotics after minor corrections.
The author needs to address the following comments/corrections.
- The author could have discussed about Cystic Fibrosis, and its symptoms with available therapies.
Author’s response: We appreciate the editor’s suggestion and amended the manuscript in line 35-41.
- The author could have shown a diagram of the disturbance of secretion of Cl- and HCO3 (epithelial Na+ channel (ENaC) dysfunction) by defective CFTR resulting mucus dehydration.
Author’s response: We appreciate the editor’s suggestion and amended the manuscript by adding the new figure 1.
- Follow the same pattern of discussion for S. aureus as in case of P. aeruginosa.
3.1.1. Prevalence
3.1.2. Virulence factors
3.1.3. Adaptation & Persistence
Author’s response: We appreciate the editor’s suggestion and amended the manuscript in line 141-145.
- The author could talk about different types of nebulizers in the introduction.
Author’s response: We appreciate the editor’s suggestion and amended the manuscript in line 53-59.
- Space missing for “300mg/5mL” (correct all).
Author’s response: We appreciate the editor’s suggestion and amended the manuscript.
- The author could discuss a bit the mechanism of action of inhaled antibiotics targeting chronic P.aeruginosa and S. aureus infection.
Author’s response: We appreciate the reviewer’s comments, however the exact mechanism of actions have been discussed in the section 3 as the first part of each antibiotic.
- For the section 3, follow same pattern (Mechanism of Action, Efficacy, Safety, and Indication) for all antibiotics.
Author’s response: We appreciate the editor’s suggestion and amended the manuscript in line 202-303, 666-672.
- Remove underline from sections 3.4.1, 3.4.2, 4.1 and 4.2 (335-346, 419-452).
Author’s response: We appreciate the editor’s suggestion and amended the manuscript.
- Provide data in the table for section 5 and 6.
Author’s response: We appreciate the editor’s suggestion and amended the manuscript (line 728-729, 733-735).
- The author should correct the format of references wherever needed (e.g Year Bold, Volume Italic etc).
Author’s response: We appreciate the editor’s suggestion and amended the manuscript.